# Anti-Algics in the Therapeutic Response of Breast and Urological Cancers

**DOI:** 10.3390/ijms25010468

**Published:** 2023-12-29

**Authors:** Ana Catarina Matos, João Lorigo, Inês Alexandra Marques, Ana Margarida Abrantes, Matilde Jóia-Gomes, Pedro Sa-Couto, Ana Cristina Gonçalves, Ana Valentim, Edgar Tavares-Silva, Arnaldo Figueiredo, Ana Salomé Pires, Maria Filomena Botelho

**Affiliations:** 1Coimbra Institute for Clinical and Biomedical Research (iCBR), Area of Environment, Genetics and Oncobiology (CIMAGO), Biophysics Institute, Faculty of Medicine, University of Coimbra, 3000-548 Coimbra, Portugalines.marques@student.uc.pt (I.A.M.); mabrantes@fmed.uc.pt (A.M.A.); edtavares@fmed.uc.pt (E.T.-S.); ajcfigueiredo@gmail.com (A.F.); mfbotelho@fmed.uc.pt (M.F.B.); 2Faculty of Pharmacy, University of Coimbra, 3000-548 Coimbra, Portugal; 3Center for Innovative Biomedicine and Biotechnology (CIBB), University of Coimbra, 3000-548 Coimbra, Portugal; acgoncalves@fmed.uc.pt; 4Department of Urology and Renal Transplantation, Centro Hospitalar e Universitário de Coimbra (CHUC), 3004-561 Coimbra, Portugal; joaolorigo@gmail.com; 5Clinical Academic Center of Coimbra (CACC), 3004-561 Coimbra, Portugal; anavalentim@gmail.com; 6Department of Mathematics, University of Aveiro, 3810-193 Aveiro, Portugal; matildejoia@ua.pt; 7Center for Research & Development in Mathematics and Applications (CIDMA), Department of Mathematics, University of Aveiro, 3810-193 Aveiro, Portugal; p.sa.couto@ua.pt; 8Coimbra Institute for Clinical and Biomedical Research (iCBR), Area of Environment, Genetics and Oncobiology (CIMAGO), Laboratory of Oncobiology and Hematology and University Clinics of Hematology and Oncology, Faculty of Medicine, University of Coimbra, 3000-548 Coimbra, Portugal; 9Anaesthesiology Service, Centro Hospitalar e Universitário de Coimbra (CHUC), 3004-561 Coimbra, Portugal

**Keywords:** breast cancer, bladder cancer, prostate cancer, local anesthetics, opioids, docetaxel

## Abstract

The effect of anti-algics on tumor progression and the overall survival of patients is controversial and remains unclear. Herein, we disclose the in vitro effects of the local anesthetics lidocaine, ropivacaine, and levobupivacaine on breast (MCF7), prostate (PC3, LNCaP), and bladder (TCCSUP, HT1376) cancer cell lines, both as monotherapy and in combination with standard-of-care therapeutics. Assays for cell proliferation, viability, death profile, and migration were performed. Additionally, we explored the clinical outcomes of opioid use through a cross-sectional study involving 200 metastatic prostate cancer patients. The main clinical data collected included the type of opioid therapy administered, dosage, treatment duration, disease progression, and overall survival. Results obtained demonstrate that treatment with local anesthetics has a promising selective anti-tumor effect on these types of cancer, with higher effects when associated with docetaxel. This points out the use of local anesthetics as an added value in the treatment of prostate carcinoma patients. Alternatively, chronic opioid use was correlated with reduced overall survival (*p* < 0.05) and progression-free survival (*p* < 0.05) at each treatment line in the observational study. While these results provide valuable insights, larger prospective studies are imperative to comprehensively evaluate the clinical impact of opioid analgesics in prostate cancer patients.

## 1. Introduction

Worldwide, in 2020, approximately 19.3 million new cancer cases and about 10.0 million cancer deaths were identified. In Europe, in the same year, nearly 4.4 million new cases were diagnosed and about 2 million deaths were reported [1]. Particularly, breast cancer (BC) was considered the most diagnosed type of cancer, with around 2.3 million new cases and more than 680,000 deaths detected worldwide, surpassing lung cancer. In women, this type of cancer was considered the leading cause of mortality [1]. In men, prostate cancer (PCa) was the second most frequently diagnosed type of cancer and the fifth leading cause of death [1]. Bladder cancer (bCa) was one of the 10 most diagnosed cancers, with approximately 573,000 new cases and 213,000 deaths. In men, bCa is the sixth most common cause of cancer and the ninth leading cause of death [1].

Among the cases diagnosed with this disease, one of the greatest challenges focuses on pain control associated with the tumor itself, diagnostic techniques, as well as therapeutic options. Ineffective cancer pain control has been shown to lead to an exacerbated and prolonged stress response, resulting in immunosuppression and the spreading of tumor cells [2]. Moreover, it is associated with patients’ diminished quality of life and survival time [3]. While opioid drugs are highly effective in managing pain, they present significant harm, including nausea and vomiting, hyperalgesia, postoperative opioid abuse, and deaths [4,5,6,7]. Moreover, some recent data suggest that this pharmacological class might influence tumor progression and patients’ overall survival and that this effect may be dependent on factors such as dose, duration of opioid exposure, type of opioid administration, and the opioid itself. But the results are controversial [4,8,9,10]. As a result, strategies such as multi-modal opioid-sparing approaches, namely regional anesthesia, have been developed to mitigate these risks [6].

Local anesthetics (LAs) such as lidocaine (Lid), ropivacaine (Rop), and levobupivacaine (Lev) are widely used in the health care of cancer patients, both for diagnosis and therapy, as well as a coadjuvant in pain management, especially in patients with metastatic disease. LAs block pain from the nerve endings into the central nervous system by reversibly blocking voltage-gated sodium channels (VGSC). Interestingly, the use of LAs, particularly the amide-linked LAs (Lid, Rop, and Lev), has been associated with less tumor progression, through inhibition of cell proliferation, viability, invasion, and migration. Evidence also highlights a lower presence of adverse and immunosuppressive effects of pre and postoperative pain and a reduced need for high doses of prescribed opioids [4,11]. In one study, biochemical recurrence in patients with prostate cancer was shown to be reduced in patients who received epidural anesthesia vs. opioid-based anesthesia [12]. Unfortunately, despite previous studies’ consistent conclusions on the effect of LAs on cancer cells, these still present some limitations, which prevent extrapolating a more definitive conclusion on the effect of the drugs on tumor cells and interaction with other therapeutics. In general, the duration of anesthetic application required to exert a significant anti-tumor effect is not achievable by local infiltration [11,13,14]. Moreover, each type of cancer presents a distinct response to the anesthetic, so there may be types of cancer that are particularly sensitive to these drugs. However, studies suggest that there may be a synergistic relationship between anesthetics and chemotherapeutic drugs, but studies are scarce [11,13,14].

Bringing together the controversial effects of opioids and LAs, it is crucial to understand the influence of using anti-algics on tumor progression and overall survival of patients, as well as their possible anti-tumor effects when applied in monotherapy and in combination with standard cancer therapeutics, on different types of tumor cells. Thus, two complementary studies were performed. In vitro, we evaluated the anti-tumor effects and assessed the selectivity of three common Las (lidocaine, ropivacaine, and levobupivacaine) on Pca, BC, and bCa cell lines. Bridging the gap from research to evidence-based intervention, deeper studies were made in metastatic Pca (mPCa) cell lines by assessing the potential combination of the Las with the chemotherapeutic drug docetaxel (Dtx). mPCa, particularly castrate-resistant prostate cancer (CRPC) is usually associated with a median survival of around 1–2 years [15]. Due to the metastasis pattern presented by this neoplasm, bone pain and functional limitation are among the main factors that impact the quality of life of those affected, making the use of opioid analgesics a common practice. Bridging the gap from research to evidence-based intervention in a translational approach, an observational study was conducted on patients with mPCa to evaluate the impact of opioid use on the patient’s prognosis.

## 2. Results

### 2.1. Las Present Selective Anti-Tumor Effects

The anti-tumor effect and selectivity of LAs were comparatively tested against five tumor and three non-tumor cell lines by MTT assay. Cells were treated with concentrations between 0.01 and 10 mM of Lid, Rop, and Lev. Figure 1 shows the dose–response curves obtained 48 h after treatment with the three LAs considered in this study. These curves allowed us to obtain the IC_50_ (half maximal inhibitory concentration) values presented in Table 1.

A dose-dependent anti-tumor effect was observed after 48 h of incubation for all cell lines studied. Moreover, the BC cell line MCF-7 was incubated with LAs for 24, 48, and 72 h and a time-dependent response was found since for longer exposure times, statistically significant lower IC_50_ values were obtained.

For 48 h of incubation, MCF-7 was the most sensitive tumor cell line to Lid (IC_50_ = 1.16 mM), Rop (IC_50_ = 0.55 mM), and Lev (IC_50_ = 0.16 mM). The most resistant tumor cell line to Lid was PC3 (IC_50_ = 3.17 mM) and to Rop and Lev was the LNCaP cell line (IC_50_ > 1 mM and IC_50_ = 0.56 mM, respectively). In addition, Lev presented the most anti-tumor effect, with lower IC_50_ values. Moreover, PC3 cell lines were significantly more sensitive (*p* < 0.001) to Rop and Lev, while LNCaP was significantly more sensitive (*p* < 0.001) to Lid. As shown in Figure 1 and Table 1 TCCSUP cell line was significantly more sensitive (*p* < 0.05) to Lid (IC_50_ = 1.99 mM) and Rop (IC_50_ = 0.86 mM), while HT1376 cells were significantly more sensitive (*p* < 0.05) to Lev (IC_50_ = 0.47 mM).

As observed in Figure 1 and Table 1, the IC_50_ values of MCF-7 were significantly lower (*p* < 0.001) than those obtained for the normal cell line, MCF12A, resulting in selectivity index (SI) values of 2.3, 1.6, and 2.6 for Lid, Rop, and Lev, respectively (Appendix A). Similar results were obtained when comparing the MCF-7 cell line to normal cells HaCaT (*p* = 0.002), in which the SI values of 2.2, 1.6, and 4 were obtained for Lid, Rop, and Lev, respectively.

Regarding prostate cell lines (Figure 1 and Table 1), all LAs showed to be selective to tumor cells since the IC_50_ values of RWPE-1 were significantly superior (*p* < 0.05) than those obtained for tumor cell line PC3, resulting in SI values of 1.3 for Lid and 2.2 for Lev. The IC_50_ values of RWPE-1 were also significantly superior (*p* < 0.001) to those obtained for tumor cell line LNCaP, with SI values of 2.4 and 1.5 for Lid and Lev, respectively (Appendix A).

### 2.2. LAs and Dtx Act Synergistically on Metastatic Prostate Cancer Cell Lines

The combination model allowed the determination of the IC_50_ values of Dtx when present in combination with the LAs. These values are compared with the IC_50_ values of Dtx obtained in monotherapy, i.e., in the absence of the LAs (Table 2). In general, the presence of LAs induced a significant decrease in the IC_50_ of Dtx in all the studied conditions, compared to the IC_50_ of the same drug in monotherapy. Dose reduction index (*DRI*) of Dtx, i.e., how many folds the dose of Dtx could be reduced in each combination, was also obtained. Particularly, regarding the association of Lid with Dtx, we observed a *DRI* value of 153 times for the PC3 cell line and 19 times for LNCaP cells. The association of Rop to Dtx resulted in a ten-fold dosage reduction in PC3 cells. The association of Lev to Dtx resulted in a *DRI* value of 18 on the PC3 cell line and a 22,800-fold dosage reduction on LNCaP cells (Table 2).

The combination model also allowed the determination of the combination index (*CI*, Figure 2). Results showed that Dtx acts synergistically with all LAs tested, mainly at higher concentrations of Dtx. This synergistic effect was more prominent in LNCaP cells.

### 2.3. Combined Therapies Decreased Cell Viability and Increased Cell Death by Necrosis

The effect of combined therapies on the cell viability and death profile of the PC3 cell line was assessed by flow cytometry through double labeling with AV/PI. As observed in Figure 3 and Appendix A, all associations induced a statistically significant decrease in cell viability compared to control (*p* < 0.001). Lower differences were seen in cells treated with the combination of Dtx and Lid, since it induced a decrease of viable cells to 67.5% compared to control (92.1%, *p* < 0.001) and to Lid (80.9%, *p* < 0.001) in monotherapy. Similar results were obtained in cells treated with Dtx and Rop, where it was found a decrease in viable cell population to 67.2% in comparison to control (92.1%, *p* < 0.001) or treated with Rop (78.8%, *p* < 0.01) in monotherapy. Greater differences were obtained after treatment with Dtx and Lev, since this association was able to reduce the percentage of viable cells to 65.6% compared to control (92.1%, *p* < 0.001), Lev (79.2%, *p* < 0.001), and Dtx (75.8%, *p* < 0.01) alone.

Moreover, the association of Dtx with Lid or Lev induced cell death by necrosis compared to control (*p* < 0.001). A more pronounced increment of the necrotic population was observed after treatment with Dtx and Lev, with an increase to 21.2% compared to Dtx (10.4%, *p* < 0.01) and Lev (13.0%, *p* < 0.001) in monotherapy. Whereas the association of Dtx and Rop induced cell death by late apoptosis/necrosis (12.5%) in comparison to control (2.25%, *p* < 0.05) or treated with Rop alone (4.0%, *p* < 0.05). 

### 2.4. Combined Therapies Induce Alteration in Cell Migration

The scratch assay revealed alterations in the wound area over time, as observed in Figure 4 and Figure 5. All associations in both cell lines presented differences in migration ability in comparison to the drugs in monotherapy after 48 h of incubation and in comparison to the control after 4, 24, and 48 h of incubation in the LNCap cell line and after 24 and 48 h of incubation in the PC3 cell line (*p* < 0.0001). Nevertheless, after 48 h, the risk in the control condition is completely closed, but not in the other conditions (Figure 4 and Figure 5).

Regarding the PC3 cell line, after 24 h, cells treated with the association of Dtx and Lev presented a wound closure of 5.6%, significantly lower than the wound closure obtained with cells treated with Dtx (18.5%, *p* < 0.05) or Lev (26.8%, *p* < 0.01) alone. After 48 h, all the associations revealed a statistically significant decrease in wound closure in comparison to all drugs alone (Figure 4).

Regarding the migration ability of the LNCaP cell line (Figure 5), after 4 h, only LNCaP cells treated with both associations revealed a statistically significant decrease in wound closure (Figure 5). Particularly, cells treated with the association of Dtx and Lid presented a wound closure of 5.2%, significantly lower than those obtained with cells treated with Lid (15.6%, *p* < 0.001) or Dtx alone (13.7%, *p* < 0.05). Similar results were obtained with cells treated with the association of Dtx and Lev (6%) compared to cells treated with Dtx (13.7%, *p* < 0.05) and Lev (14.2%, *p* < 0.01) alone. After 24 and 48 h of incubation, the wound closure of both associations was significantly lower than that obtained with LNCaP cells treated with drugs alone (*p* < 0.0001).

### 2.5. Clinical Outcomes of Castrate-Resistant Prostate Cancer Patients Treated with Opioids

Table 3 presents the general characteristics of the study sample. This study sample consisted of patients with a mean age of 76.2 ± 9.3 years. Of these patients, 48.6% had low-volume disease and 51.4% had high-volume disease, while 67.3% had low-risk disease and 32.7% had high-risk disease. The most common first-line treatment received by the population was Dtx, followed by abiraterone and enzalutamide (40.5%, 34%, and 21.1%, respectively). Nevertheless, 42.6% of the population reached second-line therapy, and 14.4% received third-line treatment. As a second line, 59.6%, 22.5%, and 14.6% received Dtx, enzalutamide, and abiraterone, respectively. Finally, as third-line therapy, 53.3%, 26.7%, and 13.3% of patients received cabazitaxel, radium-223, and Dtx, respectively. The progression-free survival (PFS) for each treatment line was 12.4 ± 10.5, 6.9 ± 3.5, and 5.5 ± 3.5 months, respectively. The effect size for the significant categorical variables (based on the chi-squared test presented in Table 3) varies from small to medium using Cramér’s V association measure classification (0.10–0.30 corresponds to small effect and 0.30–0.50 to medium effect size [16]. For each significant association, Cramér’s V value was: 0.177 (small) for variable CHAARTED, 0.194 (small) for variable LATITUDE, 0.270 (small) for the predictor therapeutic line, and 0.449 (medium) for death.

We identified 62 patients who were taking chronic opioids for pain treatment, with fentanyl being the most prescribed opioid (54.8%). The majority of the sample was taking strong opioids (72.6%), mostly in the form of transdermal formulations (69.3%), as described in Appendix A. The group receiving opioids showed reduced OS (*p* < 0.05) and PFS at each treatment line (*p* < 0.05).

In Table 4 we present a multivariate model that indicates a significant association between high-volume disease, opioid usage, and a diminished prognosis. This association is evident in both the univariate and multivariate analysis.

## 3. Discussion

Cancer-related pain is usually associated with decreased quality of life, and its intensity can range from mild to severe, be debilitating, and even lead to cancer progression. Hence, anti-algics have a key role in disease therapy management [3]. Our previous literature review showed that LAs may reduce direct and indirect breast cancer progression, as well as act synergistically with chemotherapeutic agents [3]. Based on these promising results, we have now proceeded to test the LAs lidocaine, ropivacaine, and levobupivacaine against different cancer types, particularly BC, bCa, and PCa. Within the tested concentration range (0.01–10 mM), all LAs presented substantial anti-tumor activity after 48 h of incubation in a dose-dependent manner. Liu et al. (2020) analyzed the response of 20 human tumor cell lines after 48 h of incubation with Lid and found that, although different cell lines show different responses, the proliferation of most of the cells decreased significantly after 48 h of incubation with 1–3 mM of Lid. Moreover, they analyzed the response of 15 human tumor cell lines after 48 h of incubation with Rop and observed a significant proliferation decrease in most of the cell lines for concentrations of 0.5 to 1 mM of Rop [17]. Lev at a concentration of 1 mM, after 48 h of incubation, decreased the proliferation of colon cancer, pCa, BC, cervical cancer, bone cancer, and liver cancer cells [18,19,20]. In the present study, the proliferation of most BC, bCa, and pCa cell lines significantly decreased after 48 h of incubation with 1–3 mM of Lid, with almost all IC_50_ values below 3 mM. Similar results were obtained after incubation with Rop or Lev, for which almost all IC_50_ values obtained were between 0.5 and 1 mM and below 0.5 mM, respectively.

Moreover, we compared the effects of Lid, Rop, and Lev on tumor and non-tumor cell proliferation and calculated the selectivity index (SI). All the SI values obtained were greater than 1, denoting a high selectivity of LAs for tumor cells. Consistent with other authors [18,21], Lev was the most effective, as well as the most selective LA used in this study. These data highlight the highly selective and anti-tumor potential of LAs in these types of cancer. 

Considering the differences between the tumor cell lines and the LAs used in our study, our results also demonstrated that the most aggressive tumor cell lines, namely PC3 and TCCSUP cells, were shown to be more sensitive to LAs than the respective less aggressive cell lines of the same tumor type, LNCaP and HT1376, respectively. Li et al. (2018) demonstrated that, using the same concentration range and incubation time, the BC triple-negative cell line, MDA-MB-231 (more aggressive), was more sensitive to LAs Lid, bupivacaine, Lev, and mepivacaine, while the luminal-A type MCF-7 cell line (less aggressive) was more sensitive to Rop and chloroprocaine [21]. This might be due to the “Celex” hypothesis, which suggests that LAs act by inhibiting the function of VGSC, and these channels are known to be overexpressed in aggressive cancer cells, predominantly Nav1.5 and Nav1.7, and accompanied by downregulation of K^+^ outward currents [22,23,24]. This can lead to membrane excitability and aggressiveness of cancer cells, particularly metastatic cancer cells [24,25]. Thus, this phenomenon might provide an opportunity to re-evaluate the VGSC function and the possible applications of these LAs.

Often, PCa tends to progress, and the standard-of-care treatment in these cases is chemotherapy with Dtx. Nevertheless, Dtx is highly associated with severe side effects such as neuropathic pain and hypersensitivity reactions. In the long term, PCa tumors tend to acquire drug resistance to Dtx [26]. In addition, metastasis increases pain in these patients, which makes it necessary to concomitantly use anti-algics. There is evidence in the literature that the combined use of these drugs and chemotherapy may be beneficial in reducing side effects and enhancing their quality of life. The approach used to study the potential of these new combined therapies was based on the drug combination model of Chou and Talalay [27,28]. Our findings revealed that all the associations tested inhibited the proliferation of mPCa cell lines in a dose-dependent manner. Moreover, based on interaction analysis using CompuSyn version 1.0 software, combination therapy showed a synergistic effect in both cell lines, with CI values less than one for almost all associations. These data highlight the potential of these associations to reduce the severe side effects and toxicity associated with Dtx chemotherapy. Other researchers also demonstrated that Lid can synergistically interact with cisplatin, 5-fluorouracil, mitomycin C, pirarubicin, or paclitaxel in BC [29], lung [30], hepatocellular [31,32], choriocarcinoma [33], melanoma [34], gastric [35], bladder [36], and esophageal [37] cancer cell lines. Moreover, Rop or bupivacaine act synergistically with cisplatin or 5-fluorouracil in BC [14], hepatocellular [31], gastric [35], and esophageal [37] cancer cell lines. Our results thus support the hypothesis that, in general, LAs act synergistically with chemotherapeutic agents.

Associations with the lowest CI values were used to proceed with the studies, namely cell migration and viability assays. LAs sensitized cancer cells to the effects of chemotherapeutic agents. In our study, in the PC3 cell line, combined therapies decreased the percentage of viable cells compared to control cells and their respective treatments with LA in monotherapy. The association of Lev with Dtx also decreased the percentage of viable cells more significantly than that obtained with Dtx treatment alone, emphasizing the synergistic effect previously demonstrated. Moreover, we observed an increase in cell death populations, predominantly the necrotic population. Many researchers have highlighted the impact of using LAs in association with chemotherapeutic agents. Ribeiro (2016) evaluated the therapeutic efficacy of Lid and mepivacaine, in situ and against metastatic oral cavity squamous cell carcinoma (OSCC) cell lines, either in monotherapy or in association with the chemotherapeutic agents, cisplatin and 5-fluorouracil. It was found that all LA associations with cisplatin or with 5-fluorouracil induced cell death more evidently than the respective treatments in monotherapy, highlighting a synergistic relationship between the two drugs [38]. This was also demonstrated in a study in esophageal cancer cells after treatment with Rop, Lid, bupivacaine, and mepivacaine in association with 5-fluorouracil and paclitaxel [37]. 

It is well documented that migration ability is one of the major phenotypic characteristics of tumor cells. In general, after 48 h of incubation with all the associations tested, we observed a markedly decreased migration of PC3 and LNCaP cells relative to control or monotherapy conditions. Furthermore, the association of Lev and Dtx significantly decreased the migration of LNCaP and PC3 cells at shorter times. The same effect was observed by other authors in another type of tumor cell. In two esophageal cancer cell lines, OE19 and SK-GT-4, mepivacaine, Rop, Lid, and bupivacaine were shown to act synergistically with paclitaxel or 5-fluorouracil in inhibiting migration compared to the inhibition obtained in monotherapy [37]. Nevertheless, as previously reported, the results exerted by the associations depend on the type of tumor cell [37]. Zheng and co-workers concluded that Rop and Lid significantly enhanced the in vitro efficacy of dacarbazine in inhibiting melanoma cell migration [39].

LAs concentration and duration required to exert a significant anti-tumor effect are unlikely to be achievable as well as effective. Nonetheless, the application of these drugs by local infiltration enables the injection of a considerable amount of LA in the area surrounding the tumor, achieving high concentrations in the mM range, as described in Table 1. In these circumstances, LAs may have high anti-tumor efficacy as well as being highly beneficial in the treatment of tumors, either as monotherapy or in association [11,13,14].

LAs indirectly affect cancer biology, including the reduction in the dose of opioid use. Opioids are immunosuppressive, which may contribute to tumor metastasis and cancer progression [40]. In our clinical investigation, we aimed to evaluate the prognostic impact of opioids on overall survival and progression-free survival in a group of patients with advanced prostate cancer. We identified 62 patients who were taking chronic opioids for pain treatment, with fentanyl being the most prescribed opioid. The majority of the sample was taking strong opioids, mostly in the form of transdermal formulations.

Our results showed that the group receiving opioids had reduced overall survival and progression-free survival. However, we acknowledge that our study has limitations, including selection bias, which may have influenced our findings. Patients who are prescribed opioids have more advanced disease or higher levels of pain, which could impact their overall prognosis.

Other studies have also reported conflicting results regarding the effects of opioids on cancer outcomes. Results obtained by Biki et al. found no significant difference in survival between patients receiving epidural anesthesia and those receiving opioid-based anesthesia for prostate cancer surgery [12]. On the other hand, a study by Zheng et al. reported that cancer-related pain and opioid requirements are associated with poor survival in advanced cancer patients [41]. These conflicting results might be due to different opioid use, receptor expression, and/or duration of use of opioids. In terms of opioid use, Zheng et al. found that postoperative opioid use was associated with poor survival of cancer patients, while intraoperative opioids did not show any effect on cancer survival [41]. Cancer cells express low levels of the μ-opioid receptor (MOR). Zylla et al. investigated the expression and the impact of MOR expression on the survival of 593 patients with mPCa and demonstrated that high levels of MOR expression requiring opioids had the worst PFS and OS [42]. The same authors also showed that greater opioid exposure is associated with inferior OS in PCa [42,43]. 

Therefore, it is important to interpret our findings in the context of the limitations of our study and consider the potential impact of selection bias. Future studies with larger sample sizes and more diverse patient populations are needed to further investigate the relationship between opioid use and cancer outcomes in patients with advanced prostate cancer.

## 4. Materials and Methods

### 4.1. Cell Culture

Human cancer cell lines of prostate (LNCaP, ATCC^®^ CRL-1740™ and PC3, ATCC^®^ CRL-1435™), breast (MCF-7, ATCC^®^ HTB-22™), bladder (TCCSUP, ATCC^®^ HTB-5™ and HT1376, ATCC^®^ CRL-1472™), as well as human normal prostate (RWPE-1, ATCC^®^ CRL-11609™), breast (MCF12A, ATCC^®^ CRL-10782™), and keratinocytes (HaCaT, ATCC^®^ PCS-200-011™) cell lines were used to conduct the experiments. TCCSUP, HT1376, MCF-7, and HaCaT cell lines were cultured and maintained in Dulbecco’s Modified Eagle’s Medium, DMEM (Sigma D5648), supplemented with 1% antibiotic, 0.25 mM sodium pyruvate, and 10 or 5% fetal bovine serum (FBS) for the HT1376 and HaCaT cell lines or TCCSUP and MCF-7 cell lines, respectively. LNCaP, PC3, and MCF12A cells were cultured and maintained in Roswell Park Memorial Institute Medium, RPMI-1640 (Sigma R4130), supplemented with 1 mM sodium pyruvate, 5% FBS, 500 ng/mL hydrocortisone, and 20 ng/mL of epidermal growth factor (EGF). RWPE-1 cell lines were cultured and maintained in keratinocyte serum-free medium, K-SFM (GIBCO 17005042), supplemented with 0.05 mg/mL bovine pituitary extract (BPE) and 5 ng/mL of EGF, according to supplier instructions. Cells were grown at 37 °C with 95% air and 5% CO_2_. All cells were routinely tested for mycoplasma contamination.

Lidocaine (Lid), ropivacaine (Rop), levobupivacaine (Lev), and docetaxel (Dtx) were supplied by Coimbra University Hospital Centre (CHUC). Table 5 provides the clinically relevant concentrations of the LAs used in this study [44]. Plasmatic concentrations correspond to LA concentrations following the regional block. First, we examined concentrations corresponding to direct local infiltration of LA to a maximum of 10 mM.

### 4.2. Metabolic Activity Assay and Selectivity Index Assessment

The anti-tumor effects of the LAs and Dtx, alone and in association, were evaluated by the MTT (3-(4,5-dimethylthiazolyl-2)2,5-diphenyltetrazolium bromide) assay. 

First, to evaluate the cytotoxic effect of LAs against BC, PCa, and bCa in monotherapy, cells were plated in 48-well plates at 8 × 10^4^ (MCF12A and LNCaP), 1 × 10^5^ (MCF7, HT1376, and PC3), and 1.2 × 10^5^ (TCCSUP) cells per milliliter, and incubated with increasing concentrations of the drugs, particularly Lid (0.3–10 mM), Rop (0.01–1 mM), and Lev (0.01–1 mM) for 48 h. Then, the effects of Dtx as well as combination studies were evaluated on mPCa cell lines, LNCaP and PC3, with increasing concentrations of Dtx (0.01–100 nM) for 48 h.

After this time, cell proliferation was measured and analyzed according to the protocol described by Almeida-Ferreira et al. [45] to access IC_50_ values. Results are expressed as the percentage of cell proliferation correlated with control experiments (non-treated cells). The IC_50_ values were determined through a sigmoid fitting.

The selectivity of the LAs can be quantified by their selectivity index (*SI*) value, calculated by Equation (1) as suggested by Badisa et al. [46]. An *SI* value > 1 indicates a drug is selective, while an *SI* value > 2 indicates a drug is highly selective to cancer cells.
(1)$$SI=\frac{{IC}_{50}^{non-tumor\;cell}}{{IC}_{50}^{tumor\;cell}}$$

### 4.3. Combination Studies

To evaluate the effect of the LAs combined with Dtx and ascertain the existence of synergy, the model proposed by Chou and Talalay (2005) was followed. This model aims to quantify the types of interactions between two drugs and classify those as additive, antagonistic, or synergistic [27,28]. This method requires the use of the IC_50_ values determined for each tested drug individually to obtain a new IC_50_ value for the association. In this study, all the associations of LAs with Dtx were made by treating tumor cells with a fixed concentration of LA concomitantly with increasing concentrations of Dtx. In the case of Dtx, the same range of concentrations tested in monotherapy was used. In the case of LA, the fixed concentration corresponded to the IC_50_ value obtained for each LA in monotherapy.

According to Chou et al. (2010), the basic equation of this method is derived from the unified theory of the Michaelis–Menten, Hill, Henderson–Hasselbalch, and Scatchard equations and is called the median effects equation [28,47]. The combination index (*CI*) is calculated by using Equation (2).
(2)$$CI=\frac{{\left(D\right)}_{1}}{{\left(Dx\right)}_{1}}+\frac{{\left(D\right)}_{2}}{{\left(Dx\right)}_{2}}=\frac{{\left(D\right)}_{1}}{{\left(Dm\right)}_{1}{\left[f_{a}/(1-f_{a})\right]}^{1/m_{1}}}+\frac{{\left(D\right)}_{2}}{{\left(Dm\right)}_{2}{\left[f_{a}/(1-f_{a})\right]}^{1/m_{2}}}$$
where *(D)*_1_ and *(D*)_2_ represent the concentrations of drugs 1 and 2 used in the combination, *(Dx)*_1_ and *(Dx)*_2_ are the respective theoretical values of drugs that would be necessary to achieve the response obtained experimentally, *f_a_* is the fraction of affected cells, *f_u_* is the fraction of unaffected cells, *Dm* is the median effect dose, and *m* is the sigmoidicity of the dose–effect curve. This experimental design also allowed to determine the combination index (*CI*) and infer whether the *CI* was synergistic (*CI* < 1), additive (*CI* = 1), or antagonistic (*CI* > 1).

The dose reduction index (*DRI*) was calculated using Equation (3), measuring how many folds the dose of Dtx could be reduced in the combination compared with the monotherapy.
(3)$$DRI=\frac{{IC}_{50}^{\;}\;of\;Dtx\;in\;monotherapy}{{IC}_{50}^{\;}\;of\;Dtx\;in\;combination}$$

### 4.4. Cell Viability and Cell Death Profile Analisys

The effects of the most synergistic combinations (*CI* < 1) were assessed in terms of cell migration, viability, and cell death profiles of PCa cells. LNCaP and PC3 were tested for different conditions: control (non-treated cells), IC_50_ value of each LA alone, Dtx concentration to which the minor *CI* value was obtained, and the respective combination of the LA and Dtx, according to Table 6.

Cell viability and types of cell death were assessed through annexin-V/propidium iodide (AV/PI) incorporation assay, as described by Tavares-da-Silva et al. [48]. Upon 48 h of incubation, according to the conditions described in Table 6 cells were detached, and 1 × 10^6^ cells were centrifuged for 5 min at 2500 rpm. Cells were then washed with PBS and incubated with 100 µL of binding buffer (0.01 M Hepes (Sigma H7523), 0.14 M NaCl, and 0.25 mM CaCl2 (Sigma C4901)), 2.5 µL of AV-FITC (Immunostep, ANXVKF), and 1 µL of PI (Immunostep, ANXVKF) for 15 min at 37 ◦C, in the dark. After incubation, 400 µL of binding buffer was added, and analysis was performed on a flow cytometer (FACSCalibur, Becton Dickinson, San Jose, CA, USA), using the excitation wavelength of 488 nm and emission wavelengths of 530 nm for the AV-FITC and 640 nm for the PI. The results are expressed as the percentage of viable (V), early apoptotic (EA), late apoptotic/necrotic (LA/N), and necrotic cells (N).

### 4.5. Migration Assay

Cell migration was assessed using a wound-healing assay. Cells were cultured in 24-well plates and allowed to grow until reaching 90% confluency. A “wound” was created by scratching the cell monolayer with a 200 µL pipette tip, and the cells were treated with each drug alone or in association. Images of wounds were taken immediately after (0 h) and at 4, 24, and 48 h. Cells were photographed at 20× amplification with a Motic AE31 microscope (Hong Kong, China) through the system Motic Image Plus 2.0 (Beijing, China). Cell migration into the detection zone was measured using the cell counting tool of the ImageJ analysis program (version 1.45 s). The results were expressed as the percentage of closed zone and normalized to the percentage of closed zone at 0 h.

### 4.6. Observational Study

After approval from the Ethical Committee of the Faculty of Medicine of the University of Coimbra and from the Ethical Committee of Coimbra University Hospital Centre, we reviewed the medical records of 165 patients with mPCa receiving anti-algics for their cancer pain who attended an oncologic urology consultation between December 2019 and December 2021. The size of this convenience sample was mainly due to logistical reasons (availability of the medical records), but it also maintained homogeneity in the oncological treatments among the patients. We excluded all patients with an age below 18 years, localized PCa, ECOG < 3, with non-oncologic painful comorbidities, opioid drug addicts, or recreational users. Patients with missing information were not included in the analysis (*n* = 64).

The following data were obtained from medical records: Age; histological tumor grade classified as ISUP 1 to 5; tumor castration resistance classified as “hormone senitive prostate cancer” (mHSPC) or “metastatic castration-resistant prostate cancer” (mCRPC); tumor volume defined by the CHAARTED trial classified as “high-volume” (visceral metastases and/or four or more bone metastases with at least one outside the vertebral column and pelvis) or “low-volume” otherwise; disease risk established by the LATITUDE trial and classified as “low-risk disease” or “high-risk disease; presence of other local therapeutic approaches classified as “Yes” or “No”; diagnostic date; death date; first, second, and third treatment line approaches; opioid prescribed; average dose of opioid in morphine’s equivalent; exposure time to opioid; administration route of opioid.

The a priori primary endpoint of this study was progression-free survival (PFS), defined as the time in months from diagnosis to progression. Overall survival (OS) was defined as the time in months from diagnosis to death from any cause.

### 4.7. Statistical Analysis

Statistical analysis was performed using GraphPad Prism (version 9.5.0, GraphPad Software, San Diego, CA, USA) and R software (version 4.1.2) together with RStudio (version 2022.07.1+554).

Each in vitro result is expressed as the mean ± standard error (mean ± SEM) of a minimum of three independent experiments performed in triplicate for MTT assay and a minimum of three independent experiments performed in duplicate for the flow cytometry and wound healing assays. Dose–response curves were obtained from MTT data through a sigmoidal fitting model that allows the determination of the IC_50_ values. Comparisons of IC_50_ values obtained for different LAs or cell lines were based on the confidence intervals (Table 2), considering the existence of statistical differences in cases where there is no overlap of confidence intervals. The combination index (CI) was calculated by CompuSyn version 1.0 software. The evaluation of the normality of distribution of the quantitative variables was carried out according to the normality test of Shapiro–Wilk. Parametric tests were used in the case of a normal distribution and non-parametric tests were used otherwise. For cell death assessment by flow cytometry, comparisons between groups were performed using one-factor ANOVA or the Kruskal–Wallis test, depending on normality. Pairwise comparisons were performed with Bonferroni’s correction. Outliers were identified using the ROUT method. A significance level of 0.05 was considered for the whole analysis.

In the observational study, summary statistics, including mean, standard deviation, median, and interquartile range, were recorded and calculated for continuous variables. Frequency counts and percentages for categorical variables were calculated. Fisher’s exact test or chi-square test was used to evaluate the association between two categorical variables. The Wilcoxon rank sum test was used to verify the association between a continuous variable and another interest variable. Recurrent event survival analysis was applied, in particular a stratified Cox model with a stratified counting process approach. Univariate models were fitted to evaluate the individual effect of independent variables on time-to-event outcome and to select the covariates of interest for the final model. A multivariable stratified Cox model was used for multivariate analysis to include important and significant covariates. Hazard ratios and a 95% confidence interval were also calculated for all significant covariates. 

## 5. Conclusions

The preclinical component of our work demonstrated that LAs present selective anti-tumor activity, which supports the potential use of LAs as a coadjuvant treatment. Moreover, the association of LAs with Dtx has a synergistic effect on mPCa cell lines. The introduction of LAs enhances Dtx efficacy in two mPCA preclinical models, which, from a translational perspective, may help reduce its side effects and toxicity. These data highlight the use of LAs as a valuable asset in the treatment of patients with mPCa that may possibly be applicable as a therapy for other malignant tumors. Further studies are required to assess the underlying mechanism of action of these associations as well as to support the anti-tumor effect in an appropriate animal model. 

Controlling severe pain symptoms remains a priority in advanced prostate cancer. Opioids continue to be a strong clinical weapon used to help these patients. Further prospective studies are needed to clarify the effect of pain and opioid use on cancer progression and survival.

## Figures and Tables

**Figure 1 ijms-25-00468-f001:**
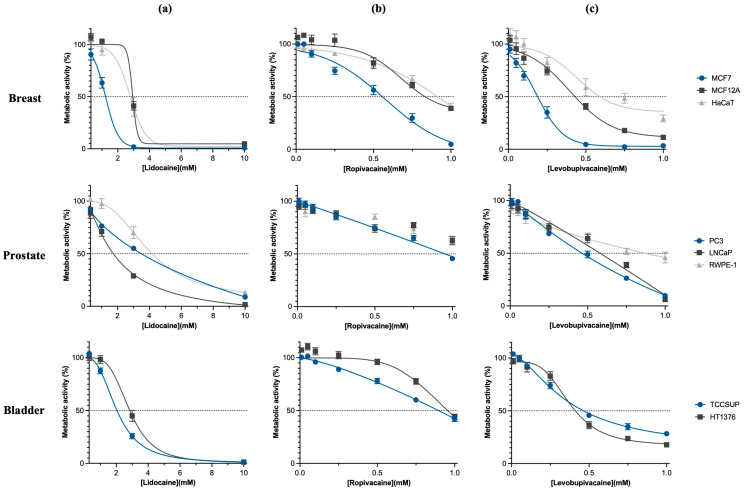
Effect of (**a**) Lid, (**b**) Rop, and (**c**) Lev on metabolic activity of human breast cancer (MCF7), prostate cancer (LNCaP and PC3), bladder cancer (TCCSUP and HT1376), and non-malignant (MCF12A, HaCaT, and RWPE-1) cell lines. Metabolic activity was evaluated as a measure of cell proliferation, by colorimetric MTT assay, 48 h after treatment with Lid (0.3–10 mM), Rop (0.01–1 mM), or Lev (0.01–1 mM). In cases where sigmoid fitting adjustment was possible, dose–response curves were obtained (R^2^ > 0.91) and the half maximal inhibitory concentrations (IC_50_) were calculated. The values are given as the mean ± SEM of, at least, three independent experiments in triplicate.

**Figure 2 ijms-25-00468-f002:**
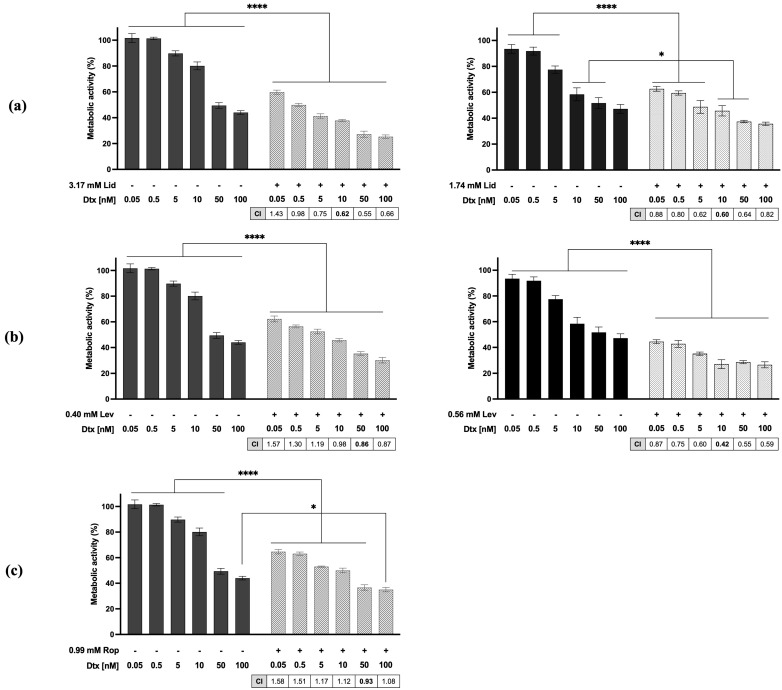
Synergistic effects of Dtx with (**a**) Lid, (**b**) Lev, and (**c**) Rop in prostate cancer cells lines PC3 (left) and LNCaP (right). Metabolic activity was evaluated as a measure of cell proliferation, by colorimetric MTT assay, 48 h after treatment. Data are mean ± SEM, *n* = 5 independent experiments; * *p* < 0.05, **** *p* < 0.0001 significantly different from control (*t*-test). Combination index (*CI*) values were obtained for 48 h of incubation time. Bold *CI* values represent the most promising combination regimen.

**Figure 3 ijms-25-00468-f003:**
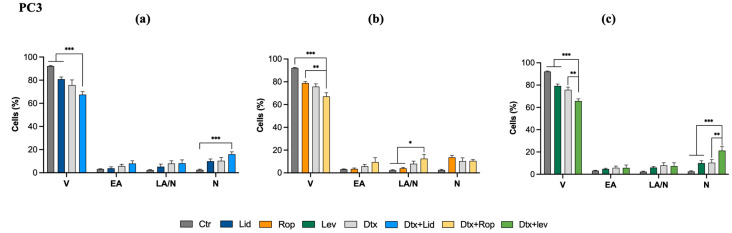
Analysis of cell viability and types of cell death induced in PC3 cells after exposure to Lid (**a**), Rop (**b**), Lev (**c**), or Dtx alone or in combined therapy for 48 h. The results are represented in percentage (%) of cell in each subpopulation: viable (V), in early apoptosis (EA), in late apoptosis/necrosis (LA/N), and in necrosis (N). The results express the mean ± SEM of, at least, three independent experiments, in duplicate. Statistically significant differences relative to control and drugs in monotherapy are marked with * *p* < 0.05, ** *p* < 0.01, and *** *p* < 0.001.

**Figure 4 ijms-25-00468-f004:**
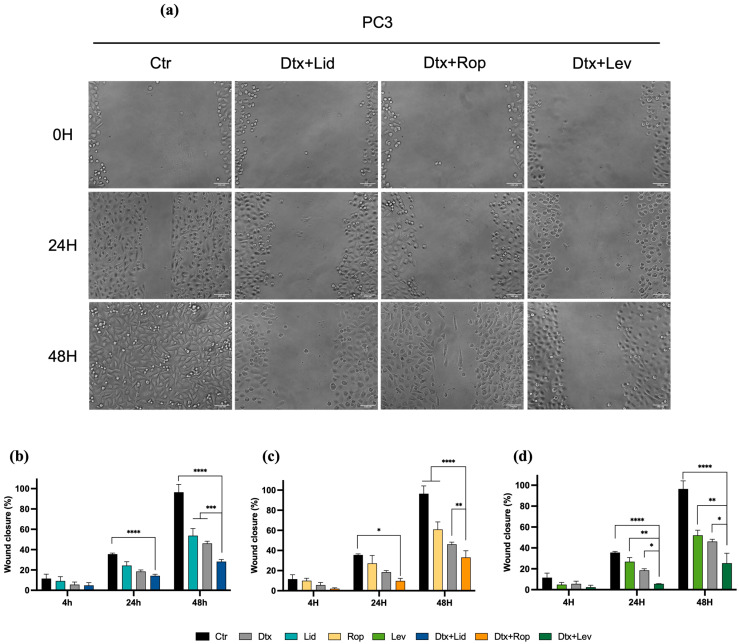
LAs inhibit PC3 cell migration. (**a**) Representative images from the wound healing assay of PC3 treated with the associations of Dtx and LAs at 20× magnification. (**b**–**d**) Effect of (**b**) Lid, (**c**) Rop, or (**d**) Lev alone or in combined therapy after 4, 24, and 48 h of incubation. The results express the mean ± SEM of at least three independent experiments, in duplicate. Statistically significant differences relative to the drugs in monotherapy are marked with * *p* < 0.05, ** *p* < 0.01, *** *p* < 0.001, and **** *p* < 0.0001.

**Figure 5 ijms-25-00468-f005:**
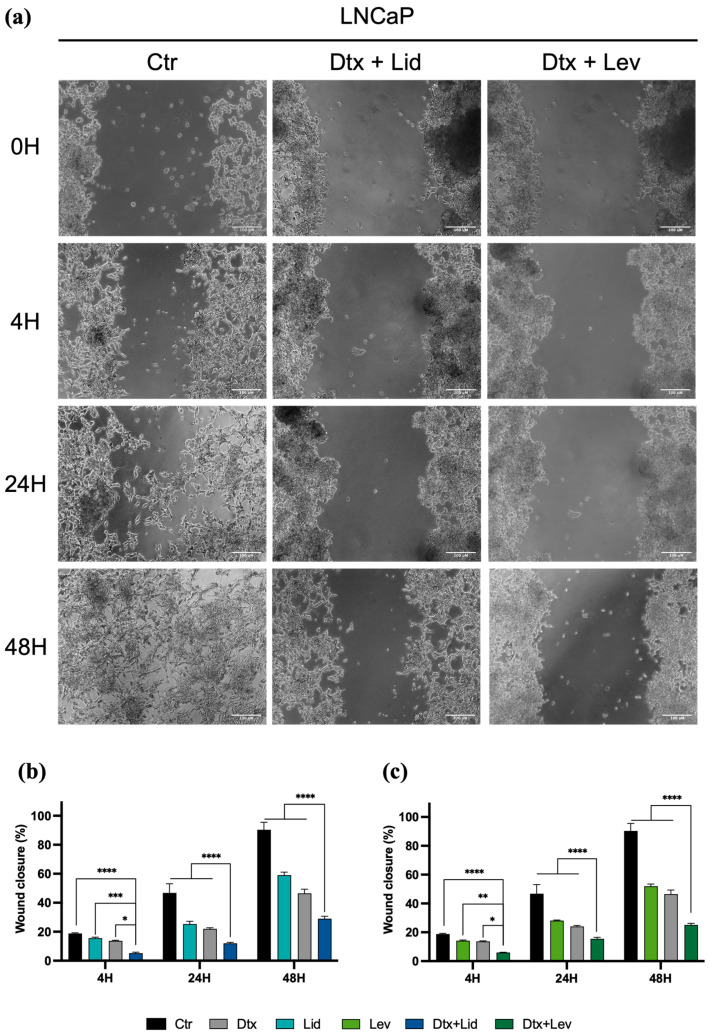
LAs inhibit LNCaP cell migration. (**a**) Representative images from the wound healing assay of PC3 treated with the associations of Dtx and LAs at 20× magnification. (**b**,**c**) Effect of (**b**) Lid or (**c**) Lev alone or in combined therapy after 4, 24, and 48 h of incubation. The results express the mean ± SEM of at least three independent experiments, in duplicate. Statistically significant differences relative to the drugs in monotherapy are marked with * *p* < 0.05, ** *p* < 0.01, *** *p* < 0.001, and **** *p* < 0.0001.

**Table 1 ijms-25-00468-t001:** IC_50_ values obtained for all cell lines after exposure to Lid, Rop, and Lev. The data express the results of, at least, three independent experiments per cell line in triplicate. Half maximal inhibitory concentrations (IC_50_) were obtained after sigmoid fitting. R^2^ is the coefficient of determination of the fitted curves. For statistical comparison, IC_50_ and 95% confidence interval (95% CoI) values are given.

	Time (h)	Lidocaine			Ropivacaine			Levobupivacaine		
		IC_50_ (mM)	R^2^	95% CoI (mM)	IC_50_ (mM)	R^2^	95% CoI (mM)	IC_50_ (mM)	R^2^	95% CoI (mM)
**Human cancer cell lines**										
MCF7	24	1.87	0.94	[1.67; 2.11]	0.90	0.96	[0.84; 0.96]	0.35	0.88	[0.31; 0.39]
	48	1.16	0.99	[0.99; 1.34]	0.55	0.97	[0.47; 0.60]	0.16	0.91	[0.14; 0.19]
	72	0.91	0.97	[0.84; 0.98]	0.42	0.84	[0.37; 0.46]	0.11	0.88	[0.10; 0.13]
LNCaP	48	1.74	0.99	[1.48; 2.06]	^a^			0.56	0.95	[0.51; 0.61]
PC3	48	3.17	0.98	[2.90; 3.50]	0.99	0.98	[0.89; 1.15]	0.40	0.97	[0.37; 0.44]
HT1376	48	1.99	0.98	[1.81; 2.18]	0.86	0.99	[0.81; 0.92]	0.47	0.99	[0.46; 0.54]
TCCSUP	48	1.84	0.99	[2.82; 2.85]	0.99	0.99	[0.94; 1.06]	0.39	0.97	[0.36; 0.42]
**Human normal cell lines**										
RWPE-1	48	4.25	0.99	[4.02; 4.72]	^a^			0.86	0.99	[0.76; 1.06]
MCF12A	48	2.69	0.99	[2.60; 2.99]	0.86	0.99	[0.82; 0.90]	0.42	0.98	[0.36; 0.45]
HaCaT	48	2.55	0.99	[2.21; 3.08]	0.90	0.98	[0.84; 0.98]	0.64	0.99	[0.54; 0.73]

^a^ Represents conditions for which sigmoid fitting could not be obtained.

**Table 2 ijms-25-00468-t002:** Values of Dtx IC_50_ and *DRI* obtained for PC3 and LNCaP cell lines after exposure to Dtx in monotherapy and in association with Lid, Rop, and Lev. Cell proliferation was evaluated as a measure of metabolic activity, by colorimetric MTT assay, 48 h after treatment with Dtx (0.01–100 nM). In cases where sigmoid fitting adjustment was possible, dose–response curves were obtained (R^2^ > 0.86) and the respective IC_50_ values were calculated. The values are given as the mean ± SE of at least three independent experiments in triplicate.

Drug/Combination	PC3				LNCaP			
	IC_50_ (nM)	R^2^	95% CoI (nM)	DRI	IC_50_ (nM)	R^2^	95% CoI (nM)	DRI
Dtx	82.79	0.96	[66.84; 104.7]	-	68.40	0.93	[43.60; 95.43]	-
Dtx/Lid	0.54	0.93	[0.36; 0.73]	153.31	3.59	0.93	[1.52; 7.87]	19.05
Dtx/Rop	8.50	0.86	[5.64; 12.30]	9.74	- ^1^	-	-	-
Dtx/Lev	4.61	0.88	[2.89; 7.01]	17.96	0.003	0.94	[0.0005; 0.01]	22,800

^1^ Indicates conditions for which sigmoid fitting could not be obtained.

**Table 3 ijms-25-00468-t003:** Mean (%) of sociodemographic and clinical characteristics of the study sample.

Variable	Total, N = 165	Non-Opioid Group, N = 103	Opioid Group, N = 62	*p*-Value ^a^
**Age** (Years; mean ± SD)	74.4 ± 9.6	74.0 ± 9.8	74.9 ± 9.3	NS
**CHAARTED**				<0.001
Low volume	69 (50.7%)	51 (37.5%)	18 (13.2%)	
High volume	67 (49.3%)	28 (20.6%)	39 (28.7%)	
Missing	29 (17.6%)	24 (23.3%)	5 (8.1%)	
**LATITUDE**				<0.001
Low risk	88 (67.7%)	63 (48.5%)	25 (19.2%)	
High risk	42 (32.3%)	14 (10.8%)	28 (21.5%)	
Missing	35 (21.2%)	26 (25.2%)	9 (14.5%)	
**Stage**				NS
MHSPC	48 (29.3%)	38 (23.2%)	10 (6.1%)	
MCRPC	116 (70.7%)	64 (39.0%)	52 (31.7%)	
Missing	1 (0.6%)	1 (1.0%)	0 (0.0%)	
**Histology**				NS
ISUP1	14 (10.5%)	10 (7.5%)	4 (3.0%)	
ISUP2	27 (20.3%)	19 (14.3%)	8 (6.0%)	
ISUP3	48 (36.1%)	33 (24.8%)	15 (11.3%)	
ISUP4	14 (10.6%)	7 (5.3%)	7 (5.3%)	
ISUP5	30 (22.5%)	14 (10.5%)	16 (12.0%)	
Missing	32 (19.4%)	20 (19.4%)	12 (19.4%)	
**Local Treatment**				NS
No	77 (57.5%)	43 (32.1%)	34 (25.4%)	
Yes	57 (42.5%)	40 (29.8%)	17 (12.7%)	
Missing	31 (18.8%)	20 (19.4%)	11 (17.7%)	
**Therapeutic Line**				
First	86 (57.3%)	69 (46.0%)	17 (11.3%)	<0.001
Second	41 (27.3%)	16 (10.7%)	25 (16.7%)	
Third	23 (15.3%)	6 (4.0%)	17 (11.3%)	
Missing	15 (9.1%)	12 (11.7%)	3 (4.8%)	
**Death**				<0.001
No	129 (78.2%)	98 (59.4%)	31 (18.8%)	
Yes	36 (21.8%)	5 (3.0%)	31 (18.8%)	
Missing	0 (0.0%)	0 (0.0%)	0 (0.0%)	

^a^ Wilcoxon rank sum test; Pearson’s chi-squared test; NS: not significant.

**Table 4 ijms-25-00468-t004:** Multivariate Cox proportional hazard model for poor prognosis in the overall sample.

	Univariate Analysis			Multivariate Model, *n* = 181		
Independent Variables	HR	CI (95%)	*p*-Value	HR	CI (95%)	*p*-Value
**Age**, *n* = 209	0.999	[0.983, 1.017]	0.938			
**CHAARTED**, *n* = 189						
Low-volume, *n* = 83	1	--	--	1	--	--
High-volume, *n* = 106	2.518	[1.687, 3.756]	<0.001	1.936	[1.174, 3.195]	0.010
**LATITUDE**, *n* = 184						
Low-risk, *n* = 114	1	--	--	1	--	--
High-risk, *n* = 70	1.691	[1.175, 2.435]	0.004	0.891	[0.544, 1.460]	0.647
**Stage**, *n* = 209						
CPHSM, *n* = 50	1	--	--			
CPRCM, *n* = 159	1.455	[0.831, 2.546]	0.189			
**Histology**, *n* = 180						
ISUP1, *n* = 16	1	--	--			
ISUP2, *n* = 31	0.848	[0.361, 1.992]	0.706			
ISUP3, *n* = 69	0.741	[0.356, 1.541]	0.422			
ISUP4, *n* = 21	0.835	[0.360, 1.939]	0.675			
ISUP5, *n* = 43	1.005	[0.500, 2.020]	0.988			
**Local treatment**, *n* = 188						
No, *n* = 113	1	--	--			
Yes, *n* = 75	0.809	[0.557, 1.175]	0.266			
**Opioid**, *n* = 209						
No, *n* = 108	1	--	--	1	--	--
Yes, *n* = 101	3.069	[2.034, 4.632]	<0.001	2.399	[1.467, 3.925]	<0.001
*Concordance* = 0.705 (*very good*)						

**Table 5 ijms-25-00468-t005:** The clinically relevant concentration of the LAs used in this study [44].

Drug	Plasmatic Concentration (μM)	Local Infiltration Concentration (μM)
Lidocaine	10	17,500 (0.5%)
Ropivacaine	3.5	7288 (0.2%)
Levobupivacaine	2.5	8667 (0.25%)

**Table 6 ijms-25-00468-t006:** Concentrations of LAs, lidocaine (Lid), ropivacaine (Rop), and levobupivacaine (Lev), and chemotherapeutic agent docetaxel (Dtx) used in each association for cell viability and death profile and cell migration studies.

Cell Line	PC3		LNCaP	
Condition	[LA] (mM)	[Dtx] (nM)	[LA] (mM)	[Dtx] (nM)
Control	0	0	0	0
Lid	3.17	0	1.74	0
Rop	0.99	0	^a^	^a^
Lev	0.40	0	0.56	0
Dtx	0	50	0	10
Dtx/Lid	3.17	50	1.74	10
Dtx/Rop	0.99	50	^a^	^a^
Dtx/Lev	0.40	50	0.56	10

^a^ Represents conditions for which sigmoid fitting could not be obtained.

## Data Availability

Data are contained within the article and Appendix A.

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
