# Peer review of "Anti-Algics in the Therapeutic Response of Breast and Urological Cancers"

_ijms, 2023, doi:10.3390/ijms25010468_

Round 1

Reviewer 1 Report

Comments and Suggestions for Authors

The authors have investigated the in vitro antitumor activity of local anaesthetics on prostate, bladder and breast cancer cells when applied in monotherapy and in combination with standard-of-care therapeutics as well as the clinical outcomes of opioid use on prostate cancer patients.  I would like to thank the authors for their valuable, interesting, and well written manuscript. There are some minor considerations that should be taken into consideration.

Minor:

1)     The expressions: et al., p (denoting probability value) should be italic.

2)     Please format your abstract and references according to the journal guidelines

3)      Please add the cytograms showing the Annexin/PI-stained quadrant charts representing apoptosis-necrosis assessment (could be added in the supplementary).

4)     Why the authors did not extend their research to involve in vivo experiment to put further emphasis on their in vitro results and make the study more comprehensive rather than performing the observational study on patients with mPCa to evaluate the impact of opioid use on patient’s prognosis.

Author Response

Dear Reviewer,

We thank you for the opportunity to submit our paper “Anti-algics in the therapeutic response of breast and urological cancers” and the opportunity for being revised.

We have carefully contemplated the concerns raised by the Reviewers and changed the manuscript in accordance with their suggestions. We thank the Reviewer for the learned and critical comments on our paper. We have thoroughly pondered over all the questions raised and revised our manuscript in accordance with the Reviewers’ concerns. Below you can find the specific answers to the remarks of the Reviewers written in blue.

REVIEWER 1

The authors have investigated the in vitro antitumor activity of local anaesthetics on prostate, bladder and breast cancer cells when applied in monotherapy and in combination with standard-of-care therapeutics as well as the clinical outcomes of opioid use on prostate cancer patients.  I would like to thank the authors for their valuable, interesting, and well written manuscript. There are some minor considerations that should be taken into consideration.

  • The expressions: et al., p (denoting probability value) should be italic.

Answer: We thank the typo highlight from the reviewer. The expressions have been corrected.

  • Please format your abstract and references according to the journal guidelines

Answer: Both abstract and references have been formatted in accordance with the journal guidelines.

  • Please add the cytograms showing the Annexin/PI-stained quadrant charts representing apoptosis-necrosis assessment (could be added in the supplementary).

Answer: We consider the suggestion of the reviewer very relevant, so as requested the dotplots of AV/PI staining were added to the supplementary materials as Figure S1.

  • Why the authors did not extend their research to involve in vivoexperiment to put further emphasis on their in vitro results and make the study more comprehensive rather than performing the observational study on patients with mPCa to evaluate the impact of opioid use on patient’s prognosis.

Answer: We thank the reviewer the question, which we consider very important. This work was performed in close collaboration with the clinicians, which allowed the research team to have the exceptional opportunity of conducting the observational study with a cohort of patients. These privileged conditions were seen as a priority. Although the time needed to perform the in vivo studies do not allow us to include those results in this manuscript, the research team will consider it for the future research.

Reviewer 2 Report

Comments and Suggestions for Authors

The authors presented the work titled as Anti-algics in the therapeutic response of breast and urological cancers. The overall quality sounds high and I feel there is no issue in any section from my side.

Minor questions are:

1. why the specific cell lines for both the cancers were selected for the study and are these conclusions valid for invasive and non-invasive?

2. Could it be worth to add the plot of directionality for migration in Figures 4 and 5.

3. In Table 4: As I see the authors used Wilcoxon rank sum test; Pearson ́s Chi-squared test, so, could you add or mention in the text about the type (+/-) of correlations?

Author Response

Dear Reviewer,

We thank you for the opportunity to submit our paper “Anti-algics in the therapeutic response of breast and urological cancers” and the opportunity for being revised.

We have carefully contemplated the concerns raised by the Reviewers and changed the manuscript in accordance with their suggestions. We thank the Reviewer for the learned and critical comments on our paper. We have thoroughly pondered over all the questions raised and revised our manuscript in accordance with the Reviewers’ concerns. Below you can find the specific answers to the remarks of the Reviewers written in blue.

REVIEWER 2

The authors presented the work titled as Anti-algics in the therapeutic response of breast and urological cancers. The overall quality sounds high and I feel there is no issue in any section from my side.

  • Why the specific cell lines for both the cancers were selected for the study and are these conclusions valid for invasive and non-invasive?

Answer: This is a very pertinent question. The rationale behind the selection of these cell lines was the fact that breast, bladder and prostate cancer patients are often subjected to therapy for pain control. In addition to the use of LAs as co-adjuvants in pain control, this pharmacological class is commonly used in technical procedures for both diagnosis and treatment. Regarding the diagnosis of these 3 cancers, LAs are used as adjuvants in breast and prostate biopsies, cytoscopies and biopsies of invaded lymph nodes. In these types of tumors, LAs are also used in the surgical resection of tumors and lymph nodes, as well as in association with radiotherapy. Although we cannot ensure these conclusions are valid for all invasive and non-invasive tumors, our results show that LAs induce an anti-proliferative effect in non-invasive cancers (as shown in MCF7 breast cancer cell line), but also a cytotoxic and synergistic effect in invasive cancer (as shown in PC3, metastatic prostate cancer cell line).

  • Could it be worth to add the plot of directionality for migration in Figures 4 and 5.

Answer: We thank the reviewer this pertinent suggestion. However, the images were acquired with the Motic AE31 microscope through the Motic Image Plus 2.0 system, a normal optical microscope as at the time of the experiments we didn't have the experimental conditions to do cell live imaging.

  • In Table 4: As I see the authors used Wilcoxon rank sum test; Pearson ́s Chi-squared test, so, could you add or mention in the text about the type (+/-) of correlations?

Answer: We thank the reviewer this pertinent suggestion. This very interesting point that we missed in the manuscript was now added in lines 248-253.
